# Anti-Obesity Effect of an Ethanol Extract of Cheongchunchal In Vitro and In Vivo

**DOI:** 10.3390/nu12113453

**Published:** 2020-11-11

**Authors:** Hye Won Kawk, Gun-He Nam, Myeong Jin Kim, Sang-Yong Kim, Gi No Kim, Young-Min Kim

**Affiliations:** 1Department of Biological Science and Biotechnology, College of Life Science and Nano Technology, Hannam University, Daejeon 34054, Korea; gyp03416@daum.net (H.W.K.); namgetpc@naver.com (G.-H.N.); jiny2415@naver.com (M.J.K.); 2Department of Food Science and Bio Technology, Shinansan University, Ansan 15435, Korea; ksychj@sau.ac.kr; 3Pharmtech Korea LTD., 59, Munmakgongdan-gil, Munmak-eup, Wonju-si, Gangwon-do 26362, Korea; gn4578@naver.com

**Keywords:** anti-obesity, Cheongchunchal, anthocyanin, 3T3-L1 preadipocytes, C57BL/6N model

## Abstract

Cheongchunchal (CE) is a developed crop more highly enriched in cyanidin-3-O-glucoside chloride (anthocyanin) than conventional waxy corn. Anthocyanin has been proven to have anti-oxidant, anti-inflammatory, anti-obesity, and anti-cancer effects. In this study, using high-performance liquid chromatography (HPLC), Cheongchunchal was confirmed to contain 8.99 mg/g anthocyanin. The inhibitory effect of an ethanol extract of Cheongchunchal (CE) on adipocyte differentiation was demonstrated using Oil Red O staining and a triglyceride assay. By conducting Western blotting, we also confirmed the regulatory effect of CE on adipocyte differentiation factors by assessing changes in the levels of factors that play a significant role in the differentiation of 3T3-L1 preadipocytes. A C57BL/6N mouse model of obesity was induced with a high-fat diet, and CE (400, 600, and 800 mg/kg/day) or Garcinia (245 mg/kg/day) was orally administered to verify the anti-obesity effect of CE. As a result of CE administration, the food efficiency ratio (FER), weight gain, and weight of tissues decreased. Additionally, blood biochemical changes were observed. Furthermore, the inhibitory effect of CE on adipocytes was confirmed through morphological observation and the expression of adipocyte differentiation-related factors in the liver and fat tissues. Therefore, in this study, we verified the anti-obesity effects of anthocyanin-rich CE both in vitro and in vivo.

## 1. Introduction

Obesity is defined as the accumulation and lipid filling of adipocytes that occur under conditions of nutritional imbalance or when the energy intake exceeds the energy consumption [1,2]. It causes many complications, such as a high blood pressure, diabetes, fatty liver, cardiovascular disease, and hyperlipidemia because of metabolic abnormalities. Additionally, obesity increases the possibility of developing osteoarthritis and back pain from excessive weight [3,4,5]. Many treatments to address the issue of obesity, including exercise, diet therapy, medication, and surgery, have been introduced [6]. However, although some obesity drugs in use are effective in inducing weight loss, their side effects limit their use as a treatment [7,8]. As a result, researchers are actively developing anti-obesity drugs using natural substances, since they have fewer side effects [9,10,11].

Obesity occurs when preadipocytes in the body are induced to differentiate, the cell cycle stops, and the number of mature adipocytes suddenly increases [12,13]. Recent studies have found that some nutrients, dietary fibers, and phytochemicals in plants inhibit differentiation into mature adipocytes. Consequently, products related to body fat reduction with these functional food ingredients are being developed [14,15,16]. Conventional waxy corn is a crop with a recessive gene—wx (waxy)—that has been popular as a snack in Korea for a long time. It is relatively easy to cultivate, and a stable cultivation area is being maintained because of the development of excellent, high-quality varieties of waxy corn; a strong market preference for well-being foods; and the development of various processed products [17,18]. Anthocyanins, which are known to possess anti-oxidant, anti-inflammatory, anti-obesity, and anti-cancer effects, are found in many crops, such as purple corn, black beans, and purple sweet potatoes.

Recently, a variety of anthocyanin-rich crops have been developed [19,20,21]. If conventional waxy corn were to be used to develop an anthocyanin-rich crop, it could be sold as green corn that could be directly consumed, and would therefore increase in value. The Cheongchunchal used in this study is a type of corn that was developed to contain large amounts of anthocyanins. The anti-obesity effects of various plants and compounds have been confirmed in many studies by investigating the inhibition of preadipocyte differentiation into mature adipocytes in vitro or the factors related to weight, tissue weight, and fat differentiation in in vivo animal models of obesity. Therefore, to confirm the anti-obesity effect of Cheongchunchal, which is an anthocyanin-rich variety of waxy corn, we studied the inhibition of differentiation in vitro in 3T3-L1 preadipocytes through a WST-1 assay, Oil Red O staining, triglyceride analysis, and Western blotting. Moreover, body fat suppression and obesity-related factors were investigated using weight gain, measured tissue weights, and the results of blood biochemical tests of the C57BL/6N model of obesity based on a high-fat diet, in order to show the anti-obesity effect of Cheongchunchal.

## 2. Materials and Methods

### 2.1. Reagents

The Cheongchunchal (Grant number: No. 6278) used in this experiment originated from Hongcheon, Gangwon-do, Korea, and the plant is a variety developed in Korea that is scheduled to be listed as an international plant resource. The listing information will be disclosed through future research. The Cheongchunchal cob was dried and crushed. Then, 2000 mL of 40% ethanol was added to 250 g of Cheongchunchal to block light, followed by reflux extraction at room temperature for 6 h. The extracted Cheongchunchal was concentrated under reduced pressure using a vacuum concentrator and then stored at −86 °C. To prepare an ethanol extract of Cheongchunchal (CE) at each concentration examined (200, 400, 800, and 1000 μg/mL), the extract was prepared by dissolving it in an equal volume of dimethyl sulfoxide (DMSO). The prepared samples for each concentration were stored frozen at −20 °C before use.

### 2.2. Quantitative Analysis of the Cyanidin-3-O-Glucoside Chloride (Anthocyanin) Content in the CE Using High-Performance Liquid Chromatography (HPLC)

The CE was prepared by mixing it with water (Burdick & Jackson, USA) and methanol (Burdick & Jackson) at a ratio of 30:70 (*v*/*v*) to 500 mg/L, followed by stepwise dilution to 1 mg/L. Anthocyanin was also diluted to 500 mg/L with the same solvent and used as a standard stock solution. For quantitative analysis of the anthocyanin content in CE, a standard curve was generated by diluting the anthocyanin to 0.1, 0.2, 0.5, 1, and 2 mg/L, and the analysis was repeated three times. All substances were analyzed by an injection of 10 µL of the sample into Shimadzu HPLC i-Series LC-2030 LT (SHIMADZU, Japan) and separation at a flow rate of 1.0 mL/min through a SunFireTM C-18 column (4.6 × 250 mm, 5 μm, Waters, Germany). In addition, mobile phase A consisted of 0.1% trifluoroacetic acid (Sigma-Aldrich, St. Louis, MI, USA) added to water, and mobile phase B consisted of 0.1% trifluoroacetic acid added to acetonitrile (Burdick & Jackson) (*v*/*v*). The change in the ratio of the mobile phases is shown in Table 1, and the absorbance at 520 nm was detected using deuterium (D2) lamps (SHIMADZU) over a 35-min period.

### 2.3. 3T3-L1 Preadipocyte Cell Culture

The 3T3-L1 preadipocytes were obtained from the American Type Culture Collection (ATCC, MD, USA), and Dulbecco’s modified Eagle medium (DMEM) containing 10% bovine calf serum (HyClone Laboratories Inc., Logan, UT, USA) and 1% antibiotics was used as the culture medium. The cells were cultured in an incubator under the conditions of 5% CO_2_ at 37 °C. After the cells had been suspended using Trypsin-EDTA (HyClone Laboratories Inc.) every 48 h, the cells were seeded at 1 × 10^6^ cells/mL and subcultured.

### 2.4. WST-1 Assay

The 3T3-L1 preadipocytes were seeded at 1 × 10^5^ cells/well in a 24-well plate for the cell culture. After incubation for 24 h, the cells were treated with CE at a range of concentrations (200, 400, 800, and 1000 μg/mL) for 48 h. A total of 50 μL of WST-1 solution (Daeillab, Korea) was added to each well and incubated for 2 h under the conditions of 5% CO_2_ at 37 °C, after which 100 μL of the medium treated with the WST-1 solution was dispensed into 96-well plates. Then, the absorbance at 450 nm was measured using a FLUOstar Omega (BMG Labtech, Otenberg, Germany).

### 2.5. Differentiation Induction

The 3T3-L1 preadipocytes were seeded at 1 × 10^5^ cells/mL in a 6-well plate in DMEM medium containing 10% bovine calf serum and 1% antibiotics. After the cells were determined to be confluent, 1 μM dexamethasone, 0.5 mM 3-isobutyl-1-methylxanthine, and 1 μg/mL insulin (DMI solution) were added to DMEM containing 10% fetal bovine serum and 1% antibiotics. The medium was mixed and used to treat the 3T3-L1 preadipocytes for 72 h. Then, CE (200, 400, 800, and 1000 μg/mL) and Garcinia (Gar; 200 μg/mL) were used together to treat the cells. Garcinia, which is a substance with proven anti-obesity effects, was used as a positive control in this study. Then, every 48 h, the medium used to culture the 3T3-L1 preadipocytes was replaced with medium containing 1 μg/mL of insulin with CE (200, 400, 800, and 1000 μg/mL) or Garcinia (Gar; 200 μg/mL) for 8 days. The N group did not induce adipocyte differentiation and did not treat any substances.

### 2.6. Oil Red O Staining

The 3T3-L1 preadipocytes were seeded at 1 × 10^5^ cells/mL in a 6-well plate, and adipocyte differentiation was induced by treatment with a DMI solution and CE (200, 400, 800, and 1000 μg/mL) or Garcinia (Gar; 200 μg/mL). After adipocyte differentiation, the cells were fixed with 10% formalin and treated with 60% isopropanol. The cells were then stained with an Oil Red O staining solution for 10 min. After washing with DW and photography using a phase contrast microscope (×100), the fat was extracted using 100% isopropanol, and the absorbance at 500 nm was measured.

### 2.7. Triglyceride Assay

A triglyceride quantification assay kit (Abcam, Cambridge, UK) was used to measure the amount of triglycerides that accumulated in the cells during adipocyte differentiation. The 3T3-L1 preadipocytes were seeded at 1 × 10^5^ cells/mL in a 6-well plate, and adipocyte differentiation was induced by treatment with a DMI solution and CE (200, 400, 800, and 1000 μg/mL) or Garcinia (Gar; 200 μg/mL). After the differentiation-induced cells had been homogenized using a 5% NP-40 solution, the reaction was repeated at 90 °C for 5 min and then at room temperature for 5 min to dissolve the triglycerides. The insoluble material was removed using a centrifuge. After the dissolved triglycerides were diluted 10-fold, lipase was used to treat the triglycerides for 20 min, after which a triglyceride probe and the triglyceride enzyme mix were reacted at room temperature for 60 min. The light intensity at a wavelength of 570 nm was measured using a spectrophotometer (FLUOstar Omega, BMG Labtech, Ortenberg, Germany).

### 2.8. Western Blotting

Western blotting was performed to confirm the expression of a protein that plays a key role upon the induction from preadipocytes to adipocytes. The 3T3-L1 preadipocytes were seeded at 1 × 10^5^ cells/mL in a 6-well plate, and adipocyte differentiation was induced by treatment with a DMI solution and CE (200, 400, 800, and 1000 μg/mL) or Garcinia (Gar; 200 μg/mL). A total of 150 μL of RIPA lysis buffer (ForBioKorea, Korea) containing 1× phosphatase inhibitor cocktail was added to the differentiated cells in each well to separate the proteins, and the cells were then centrifuged at 14,000 rpm for 20 min at 4 °C, after which the supernatant was removed. In the case of the tissue, it was treated with RIPA lysis buffer containing 1× phosphatase inhibitor cocktail, sonicated, and then centrifuged in the same manner to obtain the supernatant. The extracted protein was quantified by measuring the absorbance at 595 nm using a spectrophotometer (FLUOstar Omega, BMG Labtech, Ortenberg, Germany). Then, electrophoresis was performed using an 8% or 10% acrylamide gel, after which the proteins were transferred to a nitrocellulose membrane. The membrane was blocked for 2 h by treatment with 4% bovine serum albumin (BSA), after which antibodies against the following were applied at 4 °C overnight: Peroxisome proliferator-activated receptors (PPARγ); C/EBPα (Santa Cruz Biotechnology, Dallas, USA); Acetyl-CoA Carboxylase (ACC); p-AMPK; and β-actin (Cell Signaling Technology, Danvers, USA). Secondary antibodies were incubated with the membrane at 4 °C for 2 h, and the signal was observed using a UVITEC gel imaging system (Philekorea, Gyeonggi-di, Korea).

### 2.9. Experimental Animal Breeding and Diet

The experimental animals were 5-week-old male C57BL/6N mice obtained from ENVIGO (Indiana, USA). After adaptation to the environment for 1 week, six animals were randomly divided into six groups (normal-fat diet (NFD), high-fat diet (HFD), Garcinia 245 mg/kg/day (Gar), CE 400 mg/kg/day (400), CE 600 mg/kg/day (600), and CE 800 mg/kg/day (800)). The experimental animals were allowed to freely consume water and food, and mice in the NFD group consumed a normal-fat diet (10% fat kcal) purchased from Purina (Korea). Mice in the other five groups (HFD, Gar, 400, 600, and 800) consumed a high-fat diet (60% fat kcal) purchased from ENIGO. The feed composition is shown in Table 2. The specific experimental conditions of a temperature of 23 ± 2 °C, humidity of 50 ± 5%, and light/dark cycle of 12 h were maintained. Obesity was induced by supplying a high-fat diet for 9 weeks, and at the same time, Garcinia (245 mg/kg/day) and Cheongchunchal (400, 600, and 800 mg/kg/day) were administered orally every day. All animal experiments were conducted with the approval of the Hannam University Animal Experimental Ethics Committee (Daejeon, Korea).

### 2.10. Body Weight and Feed Efficiency Measurements

Body weight was periodically measured once a week to observe the condition of the animals during the experiment. Food was supplied daily, and the weight was measured once a week. After 24 h, the amount of feed remaining was measured, and the food intake was calculated as the difference between the weight of the supplied feed and the weight of the remaining feed. The food efficiency ratio (FER) was calculated by dividing the amount of weight gained by the amount of feed consumed during the experiment.

### 2.11. Blood and Tissue Collection and Analysis

Before the end of the experiment, the experimental animals were sacrificed after fasting for at least 12 h, and blood was collected from the abdominal vein after the abdomen was opened. The collected blood was reacted at room temperature for 30 min and then centrifuged at 14,000 rpm for 20 min to separate the serum. The concentrations of alanine aminotransferase (ALT), aspartate aminotransferase (AST), glucose, triglycerides, total cholesterol, low-density lipoprotein (LDL) cholesterol, and high-density lipoprotein (HDL) cholesterol in the collected serum were measured using a biochemical analyzer (AU480 Chemistry Analyzer, Beckman Coulter, USA). In addition, a rat/mouse insulin ELISA kit (Merck Millipore, Darmstadt, Germany) was used to measure the insulin in the serum. A kit (Adiponectin (mouse) total, HMW ELISA, ALPCO Diagnostics, Salem, USA) was used to measure adiponectin, and a mouse leptin ELISA kit (Merck Millipore, Darmstadt, Germany) was used to measure leptin in the serum. After blood collection, the liver, abdominal fat, kidneys, epididymal fat, visceral fat, and subcutaneous fat were extracted in that order. Parts of the abdominal fat, epididymal fat, and liver tissue samples were fixed in 10% neutral formalin for 48 h. After fixation, a paraffin block was prepared and stained with hematoxylin and eosin (H&E). The stained tissue was observed with an optical microscope at ×100. The remaining tissues were stored at −86 °C for Western blotting.

### 2.12. Statistical Analyses

The results of the experiments are expressed as the mean and standard error and were analyzed using the SPSS 21.0 program (IBM-SPSS, USA). The statistical significance of the differences between groups was analyzed using a *t*-test. All experiments were repeated three times, and the significance of the differences between groups was verified at the levels of *p* < 0.05 and *p* < 0.01.

## 3. Results

### 3.1. Qualitative Analysis of Anthocyanin in CE by HPLC

Anthocyanin and CE were analyzed through HPLC to determine the anthocyanin content in the CE (Figure 1A,B), and anthocyanin solutions at 0.1, 0.2, 0.5, 1, and 2 μg/mL were prepared as standard solutions from which a standard curve was created. The anthocyanin content in the CE was also measured (Figure 1C), which revealed that the CE contained 8.99 mg/g anthocyanin.

### 3.2. Confirmation of the Cytotoxic and Inhibitory Effects on Adipocyte Differentiation

To confirm the cytotoxicity of CE in the 3T3-L1 preadipocytes, the 3T3-L1 preadipocytes were treated with CE at different concentrations (200, 400, 800, and 1000 μg/mL) for 48 h, after which cytotoxicity was confirmed through the WST-1 assay. As a result, the CE was confirmed to exhibit no toxicity in the 3T3-L1 preadipocytes, because the cell viability was more than 90% upon treatment with CE at all concentrations (Figure 2A). The cells were treated with CE at each concentration examined (200, 400, 800, and 1000 μg/mL) or with Garcinia (Gar; 200 μg/mL) and DMI to confirm the effect of inhibiting the differentiation of adipocytes. The degree of adipocyte differentiation in each group was confirmed using Oil Red O staining, which stains hydrophobic components, such as triglycerides, cholesterol, and phospholipids. As a result, the number and size of the differentiated adipocytes were visually confirmed to be significantly decreased by CE or Gar compared to DMI (negative control) (Figure 2B). In addition, the Oil Red O staining intensity in the adipocytes from the different groups was compared. Compared to the N group, the staining intensity was significantly increased in all groups, but a comparison of the staining intensity with that of the DMI group confirmed the CE concentration-dependent reduction in staining. Among the groups, only those administered 800 and 1000 μg/mL of CE showed significantly decreased staining. This effect was most pronounced when the 1000 μg/mL CE-treated group was compared to the Gar group, which acted as the positive control group (Figure 2C). Therefore, the effect of CE on triglyceride accumulation in cells during adipocyte differentiation was confirmed. As a result, triglycerides were significantly increased in all groups except for the 1000 μg/mL CE-treated group compared to the N group, and were significantly decreased in all groups except the 200 μg/mL CE-treated group compared to the DMI group. In addition, the triglyceride content decreased, depending on the CE concentration (Figure 2D). Therefore, concentration-dependent adipocyte differentiation and the inhibitory effects of CE on triglyceride accumulation were confirmed.

### 3.3. Confirmation of CE-Induced Changes in the Expression of Proteins Related to Adipocyte Differentiation

When adipocyte differentiation was inhibited by CE (400, 800, and 1000 μg/mL) or Garcinia (Gar; 200 μg/mL) at each concentration tested, changes in the expression of the factors that play a key role in differentiation were confirmed. PPARγ, which is a factor that is mainly expressed in adipose tissue, promotes the differentiation of adipocytes, and C/EBPα expression is induced during late adipocyte differentiation. In addition, ACC is an enzyme that converts acetyl-CoA to malonyl-CoA and initiates the biosynthesis of fatty acids and triglycerides. PPARγ, C/EBPα, and p-ACC were observed to be decreased in all of the CE-treated groups compared to the DMI group, and CE exerted similar or better effects in all CE-treated groups when compared to the N and Gar groups. In addition, the level of AMPK, which catalyzes the process of ATP production, increases lipolysis in adipose tissue, and decreases fat synthesis, was increased by CE when compared to that in the DMI group, and this effect was confirmed to be concentration-dependent. CE at a concentration of 800 μg/mL or higher had similar or better effects than those observed in the N group, and the effects on the AMPK levels in the 1000 μg/mL CE-treated and Gar groups were the most similar (Figure 3). Therefore, CE was confirmed to inhibit or induce differentiation-related factors during its inhibition of adipocyte differentiation.

### 3.4. Confirmation of CE-Induced Changes in the Experimental Animals’ Body and Tissue Weights, Food Intake, and FER

The experimental animals were fed a high-fat diet for 9 weeks to induce obesity, and were simultaneously administered CE (400, 600, and 800 mg/kg/day) and Garcinia (Gar; 245 mg/kg/day) orally once a day. All experimental animals were weighed once a week. Weight gain was calculated using the difference in weight at weeks 1 and 9. The results showed increased weight gain in all groups compared to the NFD group, but weight gain was significantly decreased in all CE-treated groups compared to the HFD group (negative control). However, the positive control group (the Gar group) did not exhibit decreased weight gain compared to the HFD group. Through these findings, CE was confirmed to be more effective in suppressing weight gain than Gar. The assessment of food intake showed a decreased food intake in all groups when compared to the NFD group, and the food intake was not significantly decreased in any of the CE-treated groups when compared to the HFD group. The FERs in the CE-treated groups were higher than that in the NFD group and lower than that in the HFD group, but these differences were not significant. Similar changes in food intake and FER were observed in the Gar group. In addition, CE-induced changes in the liver, kidneys, and adipose tissue weight were confirmed. The weights of the liver and kidney tissues were increased in all CE-treated groups compared to the NFD group. However, compared to the HFD group, the 600 and 800 groups showed significantly decreased liver tissue weights, and the kidney tissue weights in the 400 and 800 groups were effectively decreased. CE was found to be more effective in reducing liver and kidney weights than Gar. The abdominal, epididymal, visceral, and subcutaneous fat weights were increased in all groups compared to the NFD group, but were decreased in all groups compared to the HFD group. However, visceral fat weight was not significantly reduced in any CE-treated group (Table 3). Therefore, these results confirm that CE is effective in suppressing body fat and weight gain.

### 3.5. Confirmation of Blood Biochemical Changes Induced by CE

After the collection of blood from all experimental animal groups and serum separation, blood biochemical changes were observed. To confirm the effect of CE on fatty liver disease, changes in ALT and AST were confirmed. Compared to the NFD group, all CE-treated groups showed increased ALT and AST levels, and both factors were decreased in all CE-treated groups compared to the HFD group, but this difference was only significant for the 400 group. Since ALT and AST levels were increased in the Gar group compared to the HFD group, CE effectively inhibited liver toxicity caused by fatty liver to a greater extent than Gar. Changes in serum glucose were observed to determine how CE affects blood glucose control, which is typically inhibited by obesity. Glucose levels were increased in all CE-treated groups compared to the NFD group, but were decreased significantly compared to those in the HFD group. Triglyceride, total cholesterol, HDL cholesterol, and LDL cholesterol levels were observed to confirm changes in the blood induced by CE. In all CE-treated groups, blood triglyceride levels were increased compared to those in the NFD group, and were decreased compared to those in the HFD group, but this difference was only significant for the 400 and 600 groups. The total cholesterol levels in all CE-treated groups were increased compared to that in the NFD group, and were significantly decreased compared to that in the HFD group. HDL cholesterol was observed to be similar in all CE-treated groups, and was significantly increased in all groups compared to the HFD group. LDL cholesterol was increased in all groups compared to the NFD group, but was significantly decreased in all CE-treated groups compared to the HFD group. Therefore, by measuring the biochemical changes in the blood, abnormal hepatotoxicity, blood sugar, and triglyceride and cholesterol levels due to obesity were found to be regulated by CE, but this effect was not concentration-dependent (Table 4).

### 3.6. Confirmation of Changes in the Serum Concentrations of Insulin, Adiponectin, and Leptin

Blood was collected from all experimental animals; the serum was separated; and changes in the levels of insulin, adiponectin, and leptin in the serum caused by CE were confirmed. Obesity induces resistance to insulin and reduces the function of pancreatic beta cells, which secrete insulin. Insulin was increased in the blood of the experimental animals in all groups compared to the NFD group, but this difference was not significant in the 600 and 800 groups. In addition, all groups showed decreased insulin compared to the HFD group (Figure 4A). Adiponectin is a protein that promotes the β-oxidation of fatty acids in muscles and inhibits fat synthesis in adipose tissue. Adiponectin was decreased in all groups compared to the NFD group, but was increased compared to the HFD group. However, this change in adiponectin levels was not significant in any of the groups (Figure 4B). Leptin is a protein secreted by adipocytes; when weight increases, leptin increases, inhibiting food intake and increasing physical activity. The assessment of the level of leptin expression showed that leptin was significantly increased in all groups, except in the Gar group, compared to the NFD group, and was significantly decreased in only the 600 group compared to the HFD group (Figure 4C). Therefore, the effect of CE against obesity was observed through the assessment of the above biomarkers. In addition, insulin resistance caused by obesity was indirectly confirmed to be alleviated by CE, thereby helping to alleviate type 2 diabetes.

### 3.7. Confirmation of Changes in Liver Morphological- and Adipocyte Differentiation-Related Factors

To assess the morphological changes due to fatty liver, the livers were collected from the experimental animals and subjected to H&E staining before the comparison. The results showed that fatty liver was mostly found in the HFD group, and that fatty liver was decreased in all groups compared to the HFD group (Figure 5A). In addition, trends in the expression of PPARγ, C/EBPα, and p-ACC in the liver tissues of each group were confirmed through Western blotting. The expression of all proteins tended to be decreased compared to the expression in the HFD group. (Figure 5B). Therefore, the inhibitory effect of CE on fatty liver was confirmed through morphological observation and the expression of adipocyte differentiation-related factors.

### 3.8. Confirmation of Changes in the Morphological- and Adipocyte Differentiation-Related Factors of Abdominal and Epididymal Fat

To assess the morphological changes in abdominal and epididymal fat, the abdominal and epididymal fat was collected from the experimental animals and compared through H&E staining. As a result, a large number of large adipocytes in both types of adipose tissue were observed in the HFD group, and the size of adipocytes was found to be decreased in all groups compared to the HFD group (Figure 6A,B). In addition, PPARγ and C/EBPα levels in the abdominal fat and epididymal adipose tissue were confirmed through Western blotting. Both factors in both types of adipose tissue were inhibited in all CE-treated groups compared to the HFD group (Figure 6C,D). Therefore, changes in the levels of adipocyte differentiation factors and a decrease in the size of the adipocytes in adipose tissue were observed to be induced by CE.

## 4. Discussion and Conclusions

Many studies focused on anti-obesity drugs and functional foods are being conducted to address the recently increasing number of obesity cases [22,23]. Additionally, many studies are being conducted to verify the effectiveness of anti-obesity drugs and functional foods with natural substances, as well as to reduce the various side effects of some commercial anti-obesity drugs [24,25,26]. This study was a basic study aimed at developing anti-obesity drugs and functional foods from natural substances. Cheongchunchal is a waxy corn variety developed to contain a large amount of anthocyanin. Previous studies have confirmed that anthocyanin has various effects, including anti-oxidant, anti-inflammatory, anti-obesity, and anti-cancer effects [27,28,29,30,31]. Therefore, we determined anthocyanin as an indicator component. However, anthocyanins are weak to light, and conditions for stable extraction are required [32]. We established an extraction method stably containing anthocyanins in the CE extraction process through this study. Currently, we are analyzing other components that affect anti-obesity in Cheongchunchal, in order to confirm a more accurate anti-obesity effect of CE. In this study, we quantitatively analyzed anthocyanin in CE using HPLC and estimated the anthocyanin content to be 8.99 mg/g. The cytotoxicity of CE in 3T3-L1 preadipocytes was confirmed by the WST-1 assay, but cytotoxicity was not found at all concentrations examined (200, 400, 800, and 1000 μg/mL). The differentiation of 3T3-L1 preadipocytes into adipocytes was induced by treating them with DMI. Simultaneously, the inhibitory effects of CE (200, 400, 800, and 1000 μg/mL) and Garcinia (Gar; 200 μg/mL) on adipocyte differentiation were examined. Using Oil Red O staining, we quantitatively evaluated the morphological properties of the cells and stained adipocytes. This study revealed that, compared to those in the DMI group, the number and size of adipocytes and the level of Oil Red O staining were decreased according to the concentration of CE. Triglycerides, which accumulate in cells during adipocyte differentiation, were also reduced according to the concentration of CE compared to their levels in the DMI group. Using Western blotting, CE was also found to inhibit differentiation by examining the levels of PPARγ, C/EBPα, ACC, and AMPK, which play a major role in controlling adipocyte differentiation. Therefore, we found that CE inhibited adipocyte differentiation by suppressing or inducing differentiation-related factors when the induction of 3T3-L1 preadipocytes into adipocytes was induced. To confirm the anti-obesity effect of CE in vivo, as well as in vitro, the C57BL/6N model mice were fed a high-fat diet to induce obesity, while CE (400, 600, and 800 mg/kg/day) or Garcinia (Gar; 245 mg/kg/day) was administered once a day. Compared to the HFD group, the groups treated with CE at all concentrations tested showed decreased weight gain. No change in the CE-induced food intake was found. Most of the time, the food intake was high in the HFD group. However, the highest food intake was observed in the NFD group. We think the cause of this is as follows: The normal-fat diet is harder to consume than the high fat diet. For this reason, while experimental animals grind feed, the feed splits and a large chunk of feed sometimes falls out of the feed container. However, as a result of calculating the food efficiency ratio, the highest values in the HFD group, and the lowest values in the NFD group, it was determined that there was no experimental error due to food intake. FER was decreased in the CE-treated groups compared to the HFD group, but the differences were not significant. Based on these results, a high-fat diet with a high caloric density increased the FER, but CE could not be confirmed to reduce the digestive absorption rate or utilization rate of some nutrients. Fatty liver is a condition in which fat makes up more than 5% of the liver’s weight and the weight of the liver is increased. Since a significant reduction in fatty liver was observed in the CE groups compared to the HFD group, we concluded that CE has an inhibitory effect against fatty liver. A significant decrease in kidney tissue weight was also observed in the CE groups compared to the HFD group. Furthermore, the weight of abdominal, epididymal, and subcutaneous fat was significantly reduced by CE. Although epididymal fat showed a CE-induced reduction in weight, it was not significant. These results confirm the effect of CE on weight loss and its fat-inhibitory effect. Compared to the HFD group, the CE groups showed decreases in ALT, AST, glucose, triglyceride, total cholesterol, and LDL cholesterol levels in the serum, but the HDL cholesterol level was increased by CE. Additionally, after the levels of insulin, adiponectin, and leptin were measured, insulin and leptin levels were found to be significantly reduced compared to those in the HFD group. Additionally, although an increase in adiponectin was observed, it was not significant. Via H&E staining, the inhibitory effect of CE on fatty liver was confirmed, as morphological changes in the liver tissues and changes in the levels of PPARγ, C/EBPα, and ACC in the liver tissues caused by CE were found. Through these results, we confirmed the inhibitory effect of CE on fatty liver by changes in the levels of differentiation-related factors. Morphological changes in abdominal and epididymal fat by CE were found through H&E staining, and PPARγ and C/EBPα levels were verified to be decreased in both types of adipose tissue. Consequently, we concluded that CE changed the levels of the differentiation-related factors in the two types of adipose tissue, thus inhibiting adipose tissue production. Based on the above results, this study showed that CE exerts an anti-obesity effect by inhibiting the adipocyte differentiation of 3T3-L1 preadipocytes, reducing the body weight and body fat weight (as confirmed using an obesity induction model), and changing the levels of obesity-related factors. Therefore, this study may serve as a basis to investigate the effectiveness and safety of CE as an anti-obesity drug and functional ingredient. Additionally, by reducing blood fat levels, CE may have a positive effect on various obesity-derived diseases.

## Figures and Tables

**Figure 1 nutrients-12-03453-f001:**
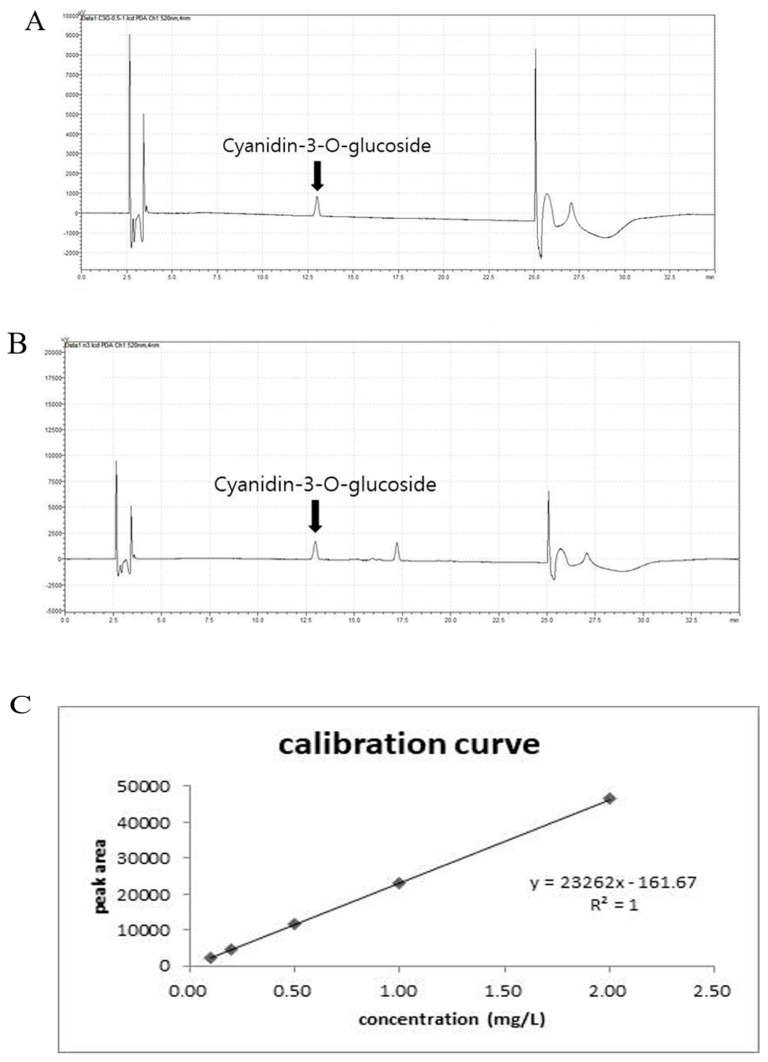
The qualitative identification of anthocyanin in Cheongchunchal (CE). (**A**) High-performance liquid chromatography (HPLC) profiles of CE at 1 mg/L. (**B**) HPLC profiles of anthocyanin at 1 mg/L. The *X*-axis represents the retention time (min), and the *Y*-axis represents the absorption units (µV). (**C**) Standard curve of anthocyanin at 0.1, 0.2, 0.5, 1, and 2 μg/mL. The detector was set at 520 nm.

**Figure 2 nutrients-12-03453-f002:**
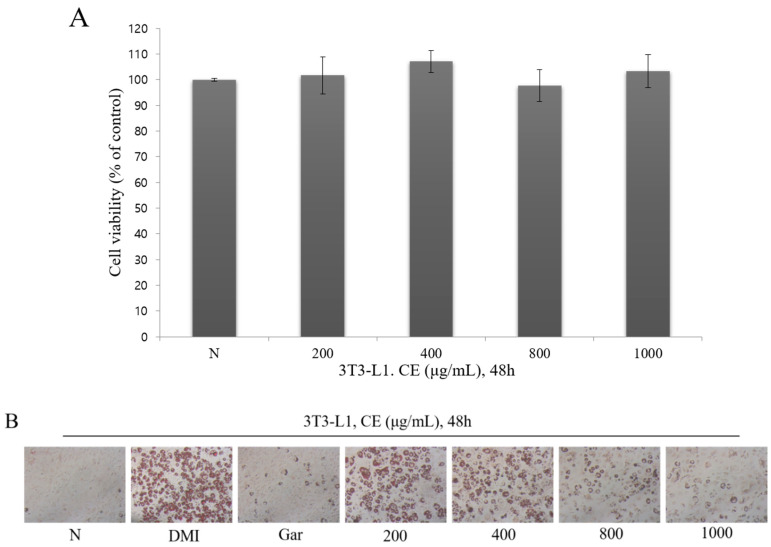
(**A**) The effect of an ethanol extract of Cheongchunchal (CE; 200, 400, 800, and 1000 μg/mL) and Garcinia (Gar; 200 μg/mL) on the cell viability of 3T3-L1 preadipocytes. Cell viability was measured by the WST-1 assay. (**B**) The number and size of differentiated adipocytes were observed with an optical microscope at ×100 (Axiovert 100, Germany). (**C**) Lipid levels (Oil Red O levels) were measured by a spectrophotometer at 450 nm. (**D**) Inhibitory effects of CE on the intracellular triglyceride in 3T3-L1 preadipocytes. The statistical analysis was carried out by the use of a *t*-test. ^b^
*p* < 0.01 compared with group not treated with any substance (N group). ^c^
*p* < 0.05 and ^d^
*p* < 0.01 compared with the 1 μM dexamethasone, 0.5 mM 3-isobutyl-1-methylxanthine, and 1 μg/mL insulin treated group (DMI group). The error bars represent the standard error.

**Figure 3 nutrients-12-03453-f003:**
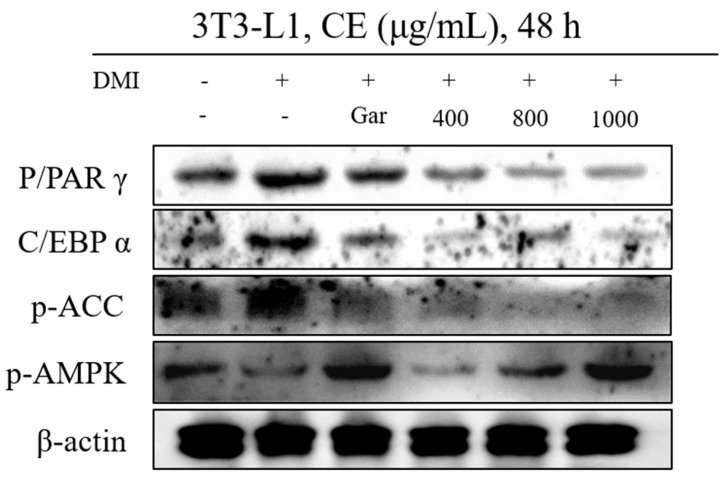
The anethanol extract of Cheongchunchal (CE; 200, 400, 800, and 1000 μg/mL) or Garcinia (Gar, 200 μg/mL) effects on the expression of adipocyte differentiation-related protein. The expression of peroxisome proliferator-activated receptors (PPARγ), C/EBPα, p-ACC, p-AMPK, and β-actin in 3T3-L1 preadipocytes was analyzed by Western blot analysis.

**Figure 4 nutrients-12-03453-f004:**
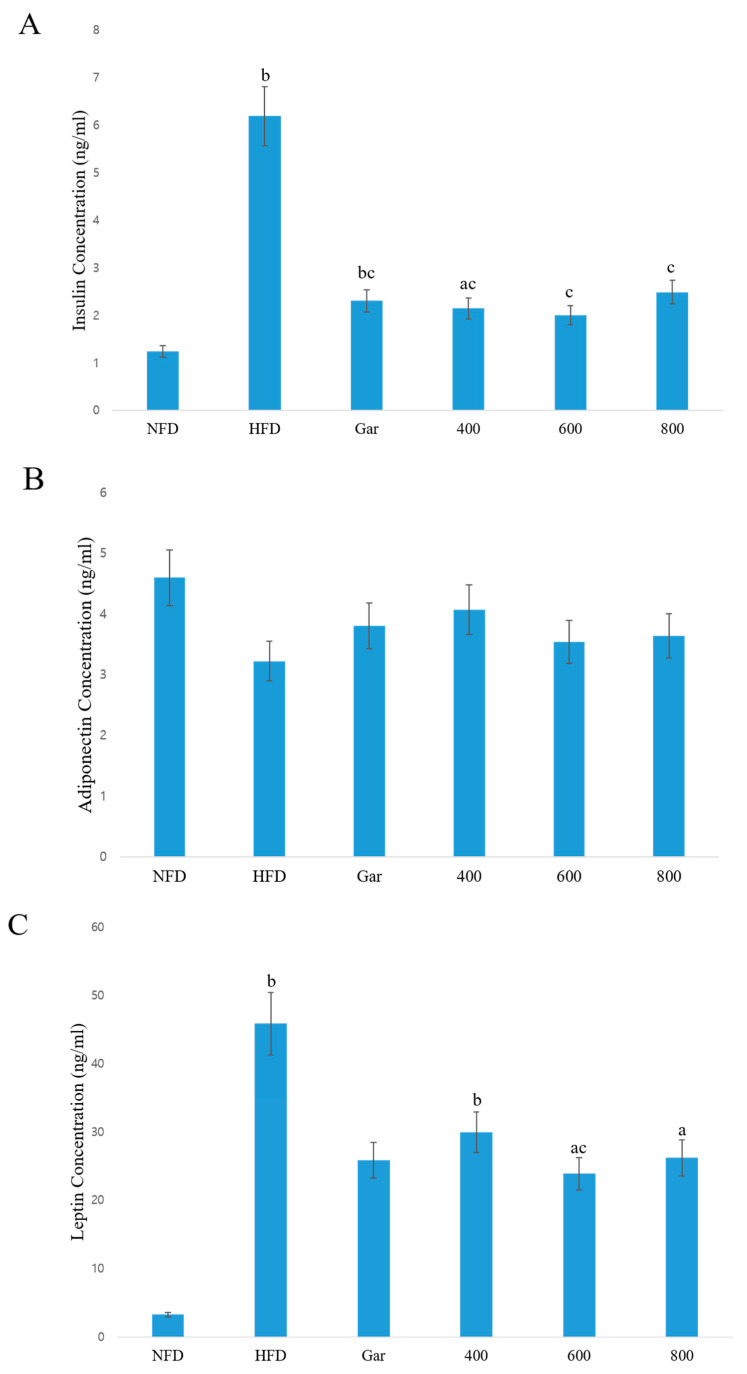
Concentrations of (**A**) insulin, (**B**) adiponectin, and (**C**) leptin in serum were measured by an ELISA kit. NFD; Normal-Fat Diet. HFD; High-Fat Diet. Gar; Garcinia 245 mg/kg/day. 400; The ethanol extract of Cheongchunchal (CE) 400 mg/kg/day. 600; CE 600 mg/kg/day. 800; CE 800 mg/kg/day. The statistical analysis was carried out by the use of a *t*-test. ^a^
*p* < 0.05 and ^b^
*p* < 0.01 compared with the NFD group. ^c^
*p* < 0.05 compared with the HFD group.

**Figure 5 nutrients-12-03453-f005:**
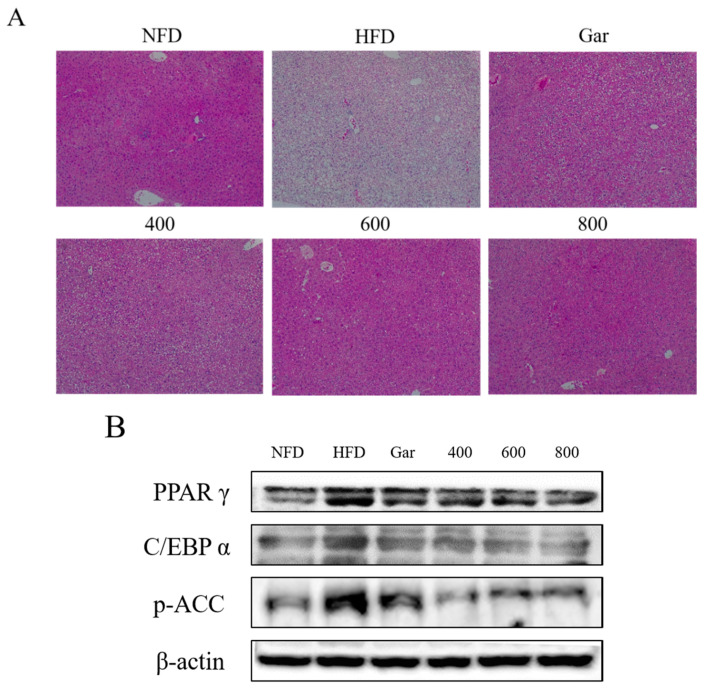
(**A**) Histological change of the liver. Hematoxylin and eosin (H&E) stained sections of liver tissues were observed with an optical microscope at ×100 (Axiovert 100, Germany). (**B**)The expression of PPARγ, C/EBPα, p-ACC, and β-actin in the liver was analyzed by Western blot analysis. NFD; Normal-Fat Diet. HFD; High-Fat Diet. Gar; Garcinia 245 mg/kg/day. 400; The ethanol extract of Cheongchunchal (CE) 400 mg/kg/day. 600; CE 600 mg/kg/day. 800; CE 800 mg/kg/day.

**Figure 6 nutrients-12-03453-f006:**
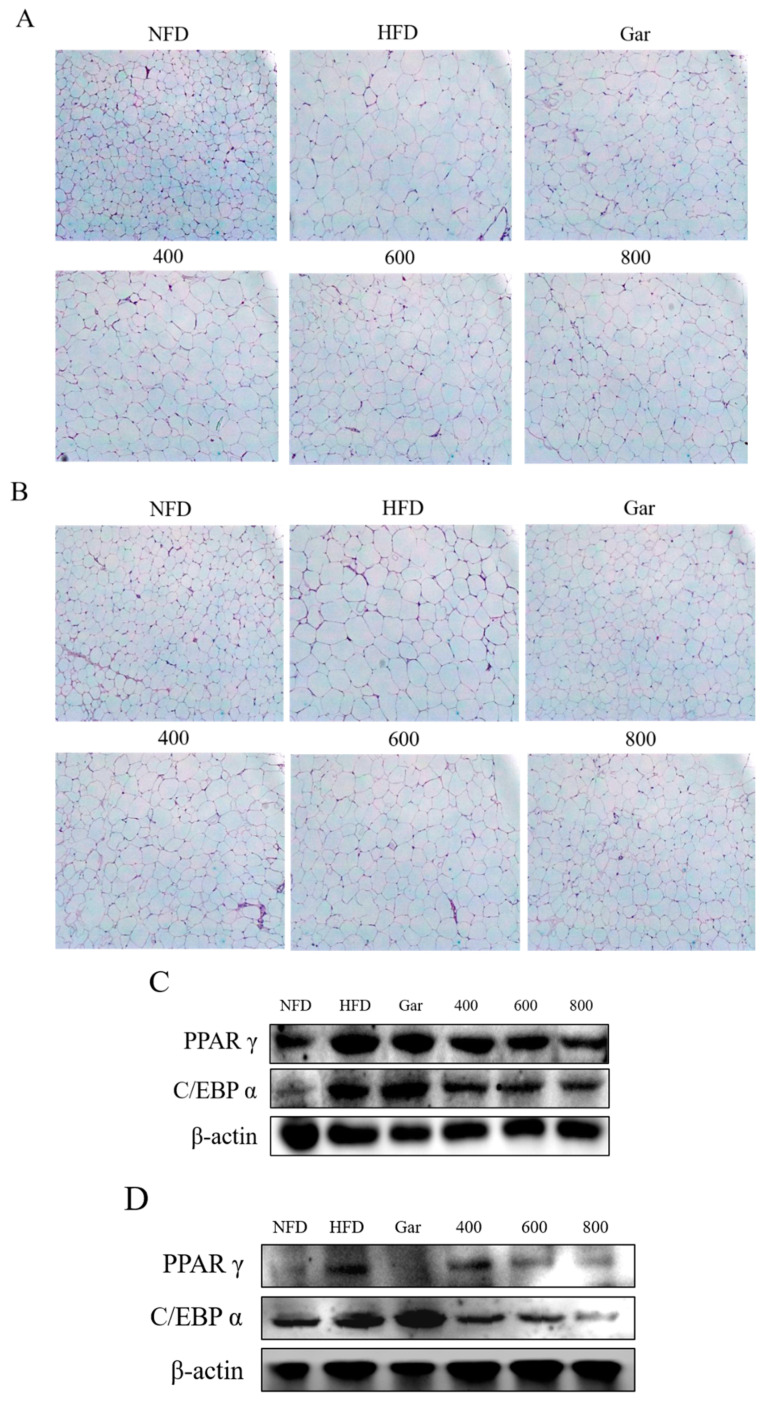
(**A**) Histological change of abdominal fat tissue. (**B**) Histological change of epididymal fat tissue. H&E stained sections of fat tissues were observed with an optical microscope at ×400 (Axiovert 100, Germany). (**C**) The expression of PPARγ, C/EBPα, and β-actin in abdominal fat tissue was analyzed by Western blot analysis. (**D**) The expression of PPARγ, C/EBPα, and β-actin in epididymal fat tissue was analyzed by Western blot analysis. NFD; Normal-Fat Diet. HFD; High-Fat Diet. Gar; Garcinia 245 mg/kg/day. 400; The ethanol extract of Cheongchunchal (CE) 400 mg/kg/day. 600; CE 600 mg/kg/day. 800; CE 800 mg/kg/day.

**Table 1 nutrients-12-03453-t001:** Change of the ratio of the mobile phase according to the retention time.

Time (min)	Mobile phase A(%) ^1^	Mobile phase B(%) ^2^
0	90	10
20	75	25
21	0	100
22	0	100
23	90	10
35	90	10

^1^ Mobile phase A (%); 0.1% trifluoroacetic acid added to water ^2^ Mobile phase B (%); 0.1% trifluoroacetic acid added to acetonitrile.

**Table 2 nutrients-12-03453-t002:** Composition of experimental diets.

Ingredient (g/kg)	Normal-Fat Diet	High-Fat Diet
Casein	200	265.0
L-cysteine	3	4.0
Corn starch	150	-
Maltodextrin	-	160.0
Sucrose	500	90.0
Cellulose	50	65.5
Soybean Oil	50	30.0
Lard	-	310.0
Mineral mixture	35	48.0
Vitamin mixture	10	21.0
Choline Bitartrate	2	3.0
Energy (kcal/g)	4	5.1
Blue Food Color	-	0.1
Protein (% kcal)	20	18.3
Carbohydrate (% kcal)	64	21.4
Fat (% kcal)	16	60.3

**Table 3 nutrients-12-03453-t003:** Measurements of weight gain, food intake, the food efficiency ratio (FER), and tissue weight.

Measurements	NFD ^1^	HFD ^2^	Gar ^3^	400 ^4^	600 ^5^	800 ^6^
Weight (g)						
1th weeks (start supplement)	21.00 ± 1.00	21.10 ± 0.75	21.50 ± 0.87	21.48 ± 0.53	21.92 ± 0.13	21.23 ± 0.78
9th weeks (end supplement)	24.63 ± 0.64	37.25 ± 2.17 ^b^	34.05 ± 1.40 ^b^	33.48 ± 0.45 ^b,c^	32.65 ± 2.31 ^b,c^	32.03 ± 1.36 ^b,c^
Weight gain (g)	3.63 ± 0.68	16.15 ± 2.44 ^b^	12.55 ± 1.69 ^b^	11.98 ± 0.94 ^b,c^	10.73 ± 2.26 ^b,c^	10.8 ± 1.47 ^b,c^
Intake						
Food intake (g/day)	3.21 ± 0.64	2.55 ± 0.43 ^a^	2.59 ± 0.63	2.32 ± 0.45 ^b^	2.26 ± 0.35 ^b^	2.27 ± 0.50 ^b^
FER^7^	0.018 ± 0.003	0.101 ± 0.02 ^b^	0.077 ± 0.01 ^b^	0.082 ± 0.01 ^b^	0.075 ± 0.02 ^b^	0.075 ± 0.01 ^b^
Tissue weight (g)						
Liver	0.93 ± 0.04	1.38 ± 0.14 ^b^	1.27 ± 0.11 ^b^	1.23 ± 0.05 ^b^	1.05 ± 0.06 ^a,d^	1.06 ± 0.08 ^a,c^
Kidney	0.31 ± 0.01	0.40 ± 0.01 ^b^	0.39 ± 0.04 ^a^	0.37 ± 0.02 ^b,c^	0.34 ± 0.02 ^d^	0.35 ± 0.01 ^b,d^
Adipose tissue weight (g)						
Abdominal fat tissue	0.385 ± 0.08	1.820 ± 0.10 ^b^	1.283 ± 0.51 ^a^	1.317 ± 0.08 ^b,d^	1.211 ± 0.24 ^b,d^	1.363 ± 0.36 ^b^
Epididymal fat tissue	0.052 ± 0.01	0.301 ± 0.04 ^b^	0.141 ± 0.05 ^a,d^	0.192 ± 0.01 ^b,d^	0.159 ± 0.03 ^b,d^	0.168 ± 0.04 ^b,d^
Visceral fat tissue	0.442 ± 0.08	1.357 ± 0.07 ^b^	1.117 ± 0.42 ^a^	1.293 ± 0.07 ^b^	1.121 ± 0.19 ^b^	1.215 ± 0.27 ^b^
Subcutaneous fat tissue	0.411 ± 0.09	2.759 ± 0.12 ^b^	1.668 ± 0.88 ^a^	1.860 ± 0.34 ^b,c^	1.201 ± 0.44 ^b,d^	1.371 ± 0.51 ^b,d^

^1^ NFD; Normal-Fat Diet. ^2^ HFD; High-Fat Diet. ^3^ Gar; Garcinia 245 mg/kg/day. ^4^ 400; The ethanol extract of Cheongchunchal (CE) 400 mg/kg/day. ^5^ 600; CE 600 mg/kg/day. ^6^ 800; CE 800 mg/kg/day. ^7^ Food Efficiency Ratio (FER) = [weight gain (g/day)]/[food intake (g/day)]. The statistical analysis was carried out by the use of a *t*-test. ^a^
*p* < 0.05 and ^b^
*p* < 0.01 compared with the NFD group. ^c^
*p* < 0.05 and ^d^
*p* < 0.01 compared with the HFD group. Values are the mean ± SD of six mice per group.

**Table 4 nutrients-12-03453-t004:** Measurements of blood biochemical changes.

Measurements	NFD ^1^	HFD ^2^	Gar ^3^	400 ^4^	600 ^5^	800 ^6^
ALT (u/L)	39.83 ± 2.78	110.97 ± 19.21 ^b^	246.13 ± 94.34 ^a,d^	60.30 ± 10.09 ^a,c^	90.27 ± 14.18 ^a^	69.03 ± 20.34
AST (u/L)	23.33 ± 0.58	116.00 ± 50.48 ^a^	136.00 ± 22.54 ^a^	37.67 ± 9.07 ^c^	76.00 ± 9.64 ^a^	75.67 ± 17.10 ^a^
Glucose (mg/dL)	220.67 ± 13.05	370.67 ± 7.57 ^b^	242.00 ± 24.25 ^d^	250.00 ± 33.06 ^d^	269.00 ± 56.40 ^c^	290.33 ± 40.80 ^a,c^
Triglyceride (mg/dL)	84.33 ± 0.58	193.67 ± 7.23 ^b^	189.67 ± 4.04 ^b^	167.33 ± 9.07 ^b,c^	151.33 ± 11.37 ^b,d^	169.00 ± 16.52 ^b^
Total-cholesterol (mg/dL)	56.67 ± 4.04	116.00 ± 18.03 ^b^	79.33 ± 5.13 ^b,c^	71.67 ± 1.53 ^b,c^	63.00 ± 3.61 ^d^	74.33 ± 9.71 ^a,c^
HDL-cholesterol (mg/dL)	109.00 ± 2.65	61.67 ± 0.58 ^b^	117.00 ± 3.61 ^a,d^	115.00 ± 7.55 ^d^	106.33 ± 5.03 ^d^	109.33 ± 3.21 ^d^
LDL-cholesterol (mg/dL)	10.33 ± 0.58	21.33 ± 1.53 ^b^	20.00 ± 2.65 ^b^	16.00 ± 2.00 ^b,c^	15.00 ± 1.73 ^a,d^	17.33 ± 0.58 ^b,c^

^1^ NFD; Normal-Fat Diet. ^2^ HFD; High-Fat Diet. ^3^ Gar; Garcinia 245 mg/kg/day. ^4^ 400; The ethanol extract of Cheongchunchal (CE) 400 mg/kg/day. ^5^ 600; CE 600 mg/kg/day. ^6^ 800; CE 800 mg/kg/day. The statistical analysis was carried out by the use of a *t*-test. ^a^
*p* < 0.05 and ^b^
*p* < 0.01 compared with the NFD group. ^c^
*p* < 0.05 and ^d^
*p* < 0.01 compared with the HFD group. Values are the mean ± SD of six mice per group.

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
