# Peer review of "Anti-Obesity Effect of an Ethanol Extract of Cheongchunchal In Vitro and In Vivo"

_nutrients, 2020, doi:10.3390/nu12113453_

Round 1

Reviewer 1 Report

The aim of this study was to verify the antiobesity effects of anthocyanin-rich Cheongchunchal extract in vitro and in vivo. Therefore, it evaluated the inhibitory effect of this extract on adipocyte differentiation, the regulatory effect on adipocyte differentiation factors by assessing changes in the levels of factors that play a significant role in the differentiation of 3T3-L1 preadipocytes. The antiobesity effect of this extract in different concentrations was comparatively tested with Garcinia by C57BL/6N mouse model of obesity induced with a high-fat diet. Furthermore, the inhibitory effect of this extract on adipocytes was confirmed through morphological observation and the expression of adipocyte differentiation related factors in liver and fat tissues. This research is important and can bring valuable information with practical application. The presented research is well-planned and the manuscript is generally well organized. There were used an appropriate and modern experimental design.

Therefore, the work could be of interest but some points have to be considered.

  • English language and style are minor spell check required.
  • In Abstract - CE abbreviation - the meaning is not specified.
  • Regarding the identification of the vegetal material, more details are needed - voucher number.
  • The phytochemical analysis of the extract could be completed with other assays. The extract may contain other components with the same action.
  • The discussions could take in attention more other literature data related with this subject, and the References could be completed.

Author Response

Thank you very much for your input.

Reviewer 2 Report

This manuscript reported that the results of basic studies of ethanol extract of Cheongchunchal (CE). Although experiments were well-designed and well-written, I think authors should make their hypothesis clear.

  1. Did authors want to show the effect of CE or anthocyanin? If authors did for CE, please explain the difference or advantage of CE comparing with other anthocyanin, such as Honeysuckle anthocyanin (Food Funct 2013, 4, 1654-1661) and blueberry anthocyanin (Int J Food Sci Nutr 2016, 67, 257-264). If authors did for anthocyanin, please explain the novel finding in authors’ current experiments.
  2. Do authors think CE is better than direct Cheongchunchal intake or direct anthocyanin supplementation? Please explain their hypothesis with the reason.

Points below are minor points.

  1. Why did authors select Garcinia as a positive control in mice study? I think other anti-obesity agents which are approved by FDA are better.
  2. Why was the food intake of HFD group reduced in this study. In general, food intake of HFD in C57BL/6 mice is increased comparing with LFD (Physiol Behav 2020, 215, 112773). Thus, the result in current study seems strange. Please explain the mechanism.
  3. Page 1, line 33 and 44; How did authors distinguish “fat cells” and “adipocytes”
  4. Page 1, line 33; Obesity is thought as not only adipocytes accumulation but also lipid filling of adipocytes. Please change authors description.
  5. Page 7, line 222; Authors described “NFD group” here. I think NFD group did not exist in 3T3-L1 experiments.
  6. Page 9, line 258; Authors described “NFD group” here. I think NFD group did not exist in 3T3-L1 experiments.
  7. I could not find the definition of N group in 3T3-L1 experiments

Author Response

Thank you very much for your input

Round 2

Reviewer 2 Report

Please add sentences below in the DISCUSSION section.

Response 4: We very much agree with your opinion. We conducted this experiment by referring to several base papers.

And most of the time, as in your opinion, food intake is high in the HFD group.

We think about the increase food intake in the NFD group:

The normal fat diet is harder than the high fat diet. For this reason, while experimental animals grind feed, the feed splits and sometimes a large chunk of feed falls out of the feed container. In this case, an inevitable error value occurs in the food intake.

We speculate that the food intake in the NFD group was high because sometimes the food dropped out of the feed container was considered food intake.

However, as a result of calculating food efficiency ratio, the highest in the HFD group and the lowest in the NFD group, so it was determined that there was no experimental error due to food intake.

Author Response

Thank you very much for your input
